

# Brief Communication: Training of AI-based nowcasting models for rainfall early warning should take into account user requirements

Georgy Ayzel[1] and Maik Heistermann[1]

[1]Institute for Environmental Sciences and Geography, University of Potsdam, Potsdam, Germany

**Abstract.** In the field of precipitation nowcasting, deep learning (DL) has emerged as an alternative to conventional tracking and extrapolation techniques. However, DL struggles to adequately predict heavy precipitation, which is essential in early warning. By taking into account specific user requirements, though, we can simplify the training task and boost predictive skill. As an example, we predict the cumulative precipitation of the next hour (instead of five minute increments), and the exceedance of thresholds (instead of numerical values). A dialogue between developers and users should identify the requirements to a nowcast, and how to consider these in model training.

## 1 Introduction

Precipitation nowcasting is the short-term prediction of where and when precipitation will occur in the immediate future, typically covering the next minutes to hours. As society becomes increasingly exposed and vulnerable to heavy rainfall, nowcasting can contribute to anticipate rapidly evolving precipitation phenomena in early warning contexts.

The standard nowcasting procedure is to track precipitation features in a series of recent radar images, and then to extrapolate their motion into the imminent future by numerical advection procedures (Germann and Zawadzki, 2002). Skillful lead times often do not exceed one hour for moderate intensities, and even less for intense convective events (Lin et al., 2024).

Over the recent years, deep learning (DL) has emerged as an alternative to conventional tracking and extrapolation techniques, starting with Shi et al. (2015), then e.g. Agrawal et al. (2019), Ayzel et al. (2020), and Ravuri et al. (2021) – followed since then by a sheer wave of new studies. The potential of DL in precipitation nowcasting lies in its capacity to discern intricate relationships in the data, without the intervention of specific feature engineering (as required for classic machine learning), or an understanding of governing processes (as required for physically-based models). The availability of massive weather radar archives in conjunction with open-source software libraries and the required computational resources (graphical and tensor processing units) provides vast opportunities for progress.

Besides some of the general issues of DL (interpretability, sensitivity to input data quality and quantity, scalability and robustness, to name a few), DL-based precipitation nowcasting struggles with the prediction of heavy precipitation features and hence extreme precipitation accumulations (e.g. Tran and Song, 2019; Ayzel et al., 2020). This is particularly frustrating since early warning is a major application scenario for nowcasting tools. Several improvements have been suggested and tested, including new architectures (Ravuri et al., 2021; Zhang et al., 2023), new types of predictive features (van Nooten et al., 2023; Leinonen et al., 2023; Kim et al., 2024), and tuning of training parameters (van Nooten et al., 2023; Franch et al., 2020). Yet, it





appears to remain difficult to successfully learn precipitation dynamics over a wide range of weather conditions, on top of the fundamental challenge to predict the spatio-temporal dynamics of convective events.

Our hypothesis is that DL models have difficulties in detecting generalizable patterns in case they are trained to predict a wide range of precipitation intensities and depths. We further hypothesize that this issue could be addressed by tailoring the training task and procedure more towards user-relevant precipitation thresholds. It is surprising that this has been rarely attempted so far (with the exception of Leinonen et al., 2023) – since the possibility to train DL models for solving specific tasks is one of their inherent strengths.

The aim of this paper is hence to demonstrate how the performance of DL models might benefit from simplifying the training task, by tailoring it more specifically towards actual user requirements. We exemplify such a simplification for two aspects:

1. **Temporal resolution of the nowcast**: typically, nowcasting models predict precipitation at temporal increments of minutes (often five minutes). This is partly historically conditioned, as the conventional numerical extrapolation schemes required a high temporal resolution for predicting the displacement of rainfall features. But while such a high resolution might be helpful for some applications, others might as well be content with anticipating the cumulative precipitation depth over the next hour. *Accordingly, we set the target variable to the precipitation depth over the next hour.*

2. **Regression vs. segmentation**: in rainfall early warning, users are not necessarily interested in the exact rainfall depth, but often rather in the exceedance of specific thresholds. The German Weather Service, for instance, uses three warning thresholds for hourly precipitation depths (15, 25, and 40 mm). Yet, the values of such thresholds can be highly context dependent. So instead of defining the training task as a regression (that aims to predict a continuous numerical variable), *we set a segmentation task in which we predict where the target variable exceeds a specific threshold.*

The starting point of our study is the Unet-based regression model RainNet (Ayzel et al., 2020, which we will here refer to as RainNet2020). RainNet2020 was shown to be superior to conventional benchmark models with regard to the prediction of low to moderate precipitation intensities; however, it even fell short to predict rainfall intensities of more than 5 mm/h. In order to provide a more competitive regression model and hence a fair experimental setup in the present study context, RainNet2020 was revised substantially: we restricted the training data to heavy rainfall events, optimised the data splitting strategy, reduced the size of the model domain, and applied some architectural improvements (see Sect. 2.3.2 for details). The resulting RainNet2024 regression model is now used as a benchmark against a set of segmentation models that operate on the same domain, with the same training and testing data and with the same architectural design – but with the training tasks set to predict the exceedance of precipitation thresholds over the next hour (instead of continuous intensities at five minute resolution).



## 2 Data and Methods

### 2.1 Precipitation data (RADKLIM)

We use the RADKLIM_YW_2017.002 dataset (Winterrath et al., 2018b, a) which is available on the open data repository of Germany's national meteorological service (Deutscher Wetterdienst; DWD hereafter). From 2001 until 2022, the dataset provides a national radar-based precipitation composite at an extent of 1100 x 900 km, a resolution of 1 km in space and 5 minutes in time. RADKLIM constitutes a consistent and homogeneous reanalysis of DWD's radar data archive, and covers comprehensive steps of quality control and corrections, including the final step of adjustment by an extended set of rain gauges.

### 2.2 Catalog of heavy rainfall events (CatRaRE)

In order to focus the model training on heavy rainfall, we used the "Catalogue of Radar-based Heavy Rainfall Events" (CatRaRE v.2021.01, Lengfeld et al., 2021a) which is openly available (Lengfeld et al., 2021b). To create this catalog, DWD extracted spatially and temporally coherent heavy rainfall objects from more than 20 years of RADKLIM data (see Sect. 2.1).

### 2.3 Nowcasting models

#### 2.3.1 RainNet2020

Being one of the first deep convolutional neural networks for radar-based precipitation nowcasting, RainNet2020 was originally published under the name "RainNet" (Ayzel et al., 2020). Its design was inspired by deep learning models from the U-Net and SegNet families. RainNet had been trained as a regression model that predicts continuous precipitation intensities on a spatial domain of 928 x 928 grid cells with a resolution of 1 x 1 km, using the summer months of 2006 to 2013 as training period. The actual target variable is the precipitation intensity at a lead time of five minutes. Nowcasts beyond that lead time are obtained in a recursive approach. In the context of this study, we use the pre-trained model exactly as it was published in 2020. It merely serves as a reference for its successor, RainNet2024.

#### 2.3.2 RainNet2024

As already pointed out in Sect. 1, we aimed to introduce a more competitive regression-type DL model which would then be consistently trained and tested together with the segmentation-type models in the context of this study. All features described in Sect. 2.3.1 for RainNet2020 also apply to RainNet2024, except for the following adjustments:

- **spatial domain**: the model is trained and applied on a spatial domain of 256 x 256 km.

- **architectural adjustments**: we used the segmentation models' library Iakubovskii (2019) as a source of model architecture. The decoder branch in the original U-Net design was substituted by the EfficientNetB4 model which balances fewer parameters with higher efficiency.





- **loss function**: we used the mean squared error (MSE) as it showed higher efficiency compared to LogCosh loss used in RainNet2020 in a number of preliminary tests.

- **training data preprocessing**: instead of data normalization by taking the natural logarithm (as implemented in Rain-Net2020 training), we used a standard linear scaling approach by dividing input data by 400 mm/h (which is close to the registered maximum intensity in the RADKLIM dataset).

Model training, validation and testing is the same as for the segmentation models (Sect. 2.3.3) and is described in Sect. 2.4.

### 2.3.3 RainNet2024-S

For predicting the exceedance of hourly precipitation thresholds, we use the very same architecture as for RainNet2024 (Sect. 2.3.2). Yet, by changing the activation function of the last linear layer from linear to sigmoid, we set it up as as a segmentation task. Accordingly, we refer to the resulting models as RainNet2024-S. Strictly speaking, the training for each precipitation threshold results into a different RainNet2024-S model. As thresholds of precipitation in the next hour, we used 5, 10, 15, 20, 25, 30, and 40 mm. The thresholds of 15, 25, and 40 mm correspond to warning levels 2 to 4 in DWD's warning protocols (DWD, 2024, in German), and should hence serve as examples of a user-specific precipitation threshold (note that warning level 1 does not exist). For RainNet2024-S training, we used the Jaccard loss function, also referred to as Intersection over Union (IoU). Jaccard loss is a relaxed and differentiable modification of the critical success index (CSI), which is a widely used metric in the field of precipitation nowcasting (Sect. 2.4).

### 2.3.4 Conventional benchmark models

We used two conventional benchmark models: the trivial "persistence" benchmark assumes that the precipitation intensities at forecast time just *persist* over the prediction lead time (in this case one hour). Considering its simplicity, though, the assumption of persistence can turn out as quite skillful. As a much more competitive benchmark, we selected PySteps (Pulkkinen et al., 2019). PySteps is a powerful open-source software tool that received a lot of attention in the recent years, and is also applied in operational contexts. It applies optical flow techniques for field tracking, and then extrapolates the detected motion into the future. In addition, PySteps allows for ensemble nowcasts that also take into account the development of the rainfall field at different scales. Here, we used PySteps in a straightforward deterministic way by using the Lucas-Kanade local feature tracking module to obtain the velocity field, which is then used to advect the latest radar image.

### 2.4 Design of benchmark experiment

The overall workflow of the benchmark experiment is summarized in Fig. 1. For model training and testing, we selected, from the CatRaRE catalog (Sect.2.2), events between 2001 and 2020 which were most extreme at a duration of six hours or less (this information is part of the catalog and is based on an analysis of the weather extremity index, see Müller and Kaspar, 2014). That way, we created a particularly challenging benchmark environment, since we not only focus our analysis on extreme precipitation events, but specifically on events with a relatively short duration. This increases the proportion of convective





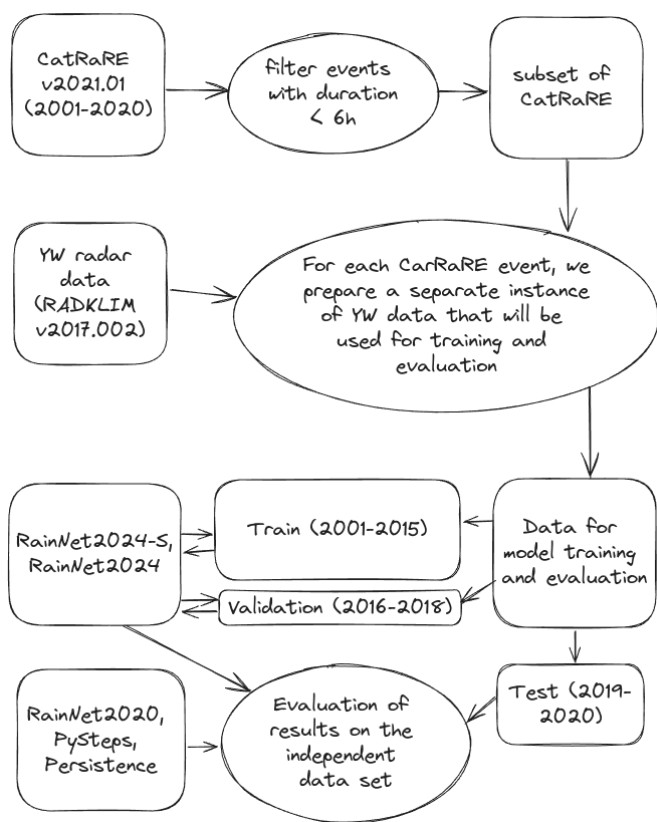

**Figure 1.** Overview of the experimental setup.

events which are, on the one hand, specifically hard to predict, but, on the other hand, constitute the kind of events that actually motivate nowcasting applications in early warning contexts.

Altogether, 19613 events were selected from CatRaRE. Using, for each event, a one hour buffer around the start and end time together with the spatial bounding box, data cubes with grid dimensions of 256 x 256 km were extracted from the RADKLIM dataset. Stacked together, these data cubes constituted the data available for training (2001-2015), validation (2016-2018)

and testing (2019-2020). For each data split and precipitation threshold (5, 10, 15, 20, 25, 30, 40 mm), we evaluated the corresponding CatRaRE events and created an index that points out the event's ID and the specific timestep of the data cube when the hourly rainfall is equal to or exceeds the threshold. For RainNet2024-S training and validation, we used only data relevant to the particular threshold exceedance while for threshold-agnostic RainNet2024, we used the full index as obtained from a threshold exceedance of 5 mm. All models were tested on the same data with regard to the particular thresholds.

For training the RainNet2024-S and RainNet2024 models, we utilized the Adam optimizer with a standard set of parameters. Both models were trained for 20 epochs. If the validation loss did not decrease for two consecutive epochs, we reduced the



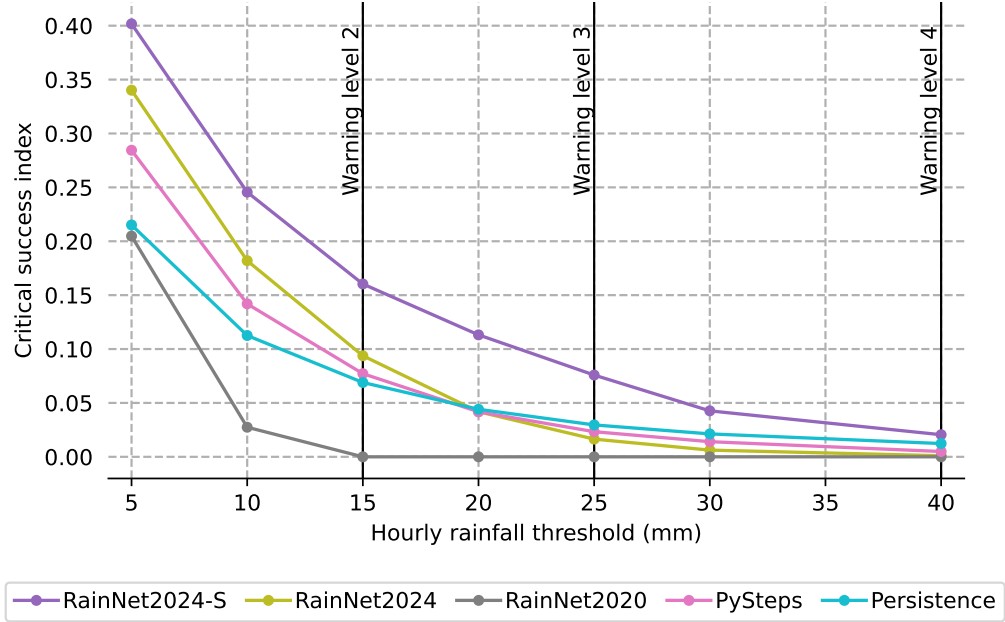

**Figure 2.** Skill of the models (in terms of CSI) in predicting the exceedance of increasingly high thresholds of precipitation depth (x-axis) that accumulate over a period of one hour after forecast time. The vertical black lines represent the DWD warning levels for hourly precipitation.

learning rate by a factor of 0.1 to refine the optimization procedure. The final models were saved in a format that preserves their configuration details (architecture) and weights, ensuring transferability and reproducibility of results.

For model testing, we used two different community-approved verification metrics (both are documented in Ayzel et al., 2020): (1) the critical success index (CSI) measures the rate of correctly forecast events relative to all forecasts except majority class hits, adjusted for random hits; (2) the fractions skill score (FSS) compares forecast and observed fractions that exceed a threshold for increasingly large neighborhoods around a pixel and hence provides a measure of how the skill changes if an increasing level of displacement error becomes acceptable.

## 3 Results and discussion

Fig. 2 presents the key results of this study. It shows the skill of the models in predicting the exceedance of increasingly high thresholds of precipitation depth that accumulated over a period of one hour after forecast time. The model skill is quantified in terms of the critical success index (CSI). Remember that the models RainNet2020, RainNet2024, PySteps and persistence predict continuous values of precipitation intensities at five minutes resolution, while the RainNet-S models were separately trained to predict threshold exceedance.

The first and, maybe, unedifying impression from Fig. 2 is that the predictive skill is moderate at best for all models, and that it strongly deteriorates with increasing precipitation thresholds (essentially no skill left at a threshold of 40 mm). Unedifying





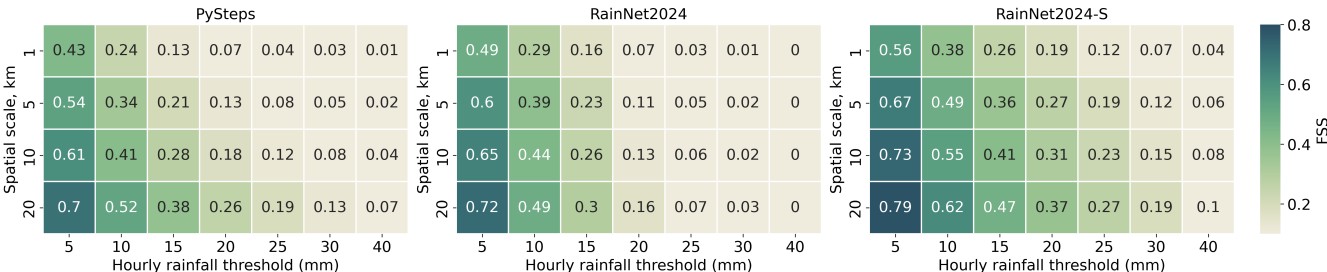

**Figure 3.** Fraction Skill Score (FSS) with increasing thresholds and spatial scales for different models

as it may be, this fact is unsurprising and well in line with the existing body of literature: high hourly precipitation depths are typically caused by convective events which are, in turn, characterised by low predictability in terms of initiation, motion and intensity dynamics. By testing the models on such events, we created an exceptionally challenging benchmark arena.

Leaving this first impression behind, though, we observe clear differences between the models. For the record, we can establish that the revision of RainNet2020 towards RainNet2024 caused a substantial boost in model skill across all precipitation thresholds, so that we can now consider RainNet2024 as a competitive benchmark: it outperforms the conventional benchmark models, PySteps and persistence, up to a precipitation depth of 15 mm in one hour (which is referred to as "warning level 2" by the DWD). For 20 mm per hour and more, both RainNet2024 and PySteps fall behind persistence, although it should be noted that the differences are as marginal as the remaining model skill at these precipitation thresholds.

    Based on Fig. 2, we can maintain that the RainNet2024-S models clearly outperform all competitors across all precipitation thresholds. The gain in the CSI metric, as compared to the corresponding second best model, is consistently around 0.06. Given the loss of skill with increasing thresholds, the relative gain in skill substantially increases with precipitation thresholds.

    These results are in line with our hypothesis that making the training task more specific pays off by a higher predictive

skill. One might argue that this result is unsurprising. In our view, though, it is by no means self-evident that the segmentation models could actually capitalize on a more specific training task.

    Fig. 3 extends the view on model skill by showing how it depends on spatial scale. It is well known that, particularly in convective situations, nowcasting models struggle to provide skillful forecasts at the kilometre-scale. The Fractions Skill Score (FSS) quantifies the model skill when we relax this requirement, i.e. when we allow an increasing level of displacement error.

Accordingly, Fig. 3 shows that the skill increases with spatial scale for all models. The RainNet2024-S model family, however, outperforms RainNet2024 and PySteps at all spatial scales and rainfall thresholds. The performance gap (i.e. the FSS difference between RainNet2024-S and its competitors) even increases with spatial scale in most of the cases (and never decreases).

    Altogether, the RainNet2024-S model family substantially outperforms all competing models at all considered thresholds, metrics and scales. The FSS demonstrates an additional dimension along which the training task for precipitation nowcasts

could be relaxed in case users do not require a kilometre-scale resolution. Although RainNet2024-S is already superior at all spatial scales, its skill might well be pushed further if directly trained for a specific spatial scale, or, in other words, if the displacement error acceptable by the user were directly considered in model training.



## 4 Conclusions

This study was motivated by the fact that DL-based models for precipitation nowcasting are still challenged by the prediction
of heavy precipitation. Our hypothesis was that they have difficulties in detecting generalizable patterns in case they are trained
to predict a wide range of precipitation intensities and depths. We further hypothesized that this issue could be addressed by
tailoring the training task and procedure more towards target variables that are actually user-relevant. That way, the training
task could be simplified, so that that the model may develop additional skill in solving it. We exemplified such a simplification
by relaxing two requirements: (i) instead of predicting rainfall intensities in five minute increments over the next hour (as
typically done in the nowcasting community), we set the target variable directly as the cumulative precipitation depth over the
next hour; (ii) instead of predicting continuous precipitation values, we trained to predict the exceedance of specific thresholds
(exemplified by DWD warning levels, but could take any other value as required by users).

To demonstrate the validity of our hypothesis, we set up a benchmark experiment in which we compared a regression-type
DL model (RainNet2024, successor of the original RainNet model published by Ayzel et al., 2020) to its segmentation-type
counterparts (RainNet2024-S). The latter were individually trained to predict the exceedance of 5, 10, 15, 20, 25, 30, and
40 mm of precipitation in the hour after forecast time. The RainNet2024-S models outperformed RainNet2024 and the other
benchmark models (PySteps, persistence) for all investigated thresholds and verification metrics.

For all models and thresholds, though, the predictive skill is still moderate to low. This is, however, also a result of the
challenging benchmark environment that was created by focusing on short-duration heavy rainfall events for training and
testing. Furthermore, we could show a substantial increase in skill for all models (but particularly for RainNet2024-S) at
spatial scales larger than the original km-resolution.

We are confident that there are, among the many new DL-based nowcasting models that were recently proposed, quite a
number of models that would outperform RainNet2024 and probably also our RainNet2024-S model family. These models
employ advanced architectures, in combination with new predictive features such as digital elevation models, polarimetric
radar moments, or fields from numerical weather prediciton models. At this point, we would like to reiterate that the aim of
our study was *not* to introduce superior DL architectures or model structures, but to demonstrate how a simplification of the
training task can help to improve model skill and to boost the usefulness for specific user groups. In our view, this approach
should be systematically explored also for recently proposed DL models.

There are various conceivable dimensions along which user preferences might find their way into model training, e.g. by
specifying precipitation thresholds, spatial and temporal resolution, or preferences towards deterministic versus probabilistic
forecasts. Our main message is hence that model developers and users need to start a dialogue of what users actually require
from a nowcast, and how this information could be effectively considered in model training.

*Code availability.* The model code together with pre-trained model weights and test data are available in the following repository: https:
//github.com/hydrogo/the-rainnet2024-family.



*Data availability.*  All data used in this study are openly available on DWD's open data server under https://opendata.dwd.de/climate_ environment/CDC: the radar-based precipitation data reanalysis RADKLIM (Winterrath et al., 2018b), as well as the CatRaRE catalog of radar-based heavy rainfall events (Lengfeld et al., 2021b).

*Author contributions.*  GA and MH conceptualized the study, GA carried out the model development and benchmark experiment, GA and MH prepared the figures, and MH wrote the manuscript with contributions of GA.

*Competing interests.*  The contact author has declared that none of the authors have any competing interests.

*Acknowledgements.*  This research was funded via the project "Innovative Instrumente zum Management des Urbanen Starkregenrisikos (InnoMAUS)" (grant no. 02WEE1632A) by the German Ministry of Education and Research (Bundesministerium für Bildung, Wissenschaft und Forschung, BMBF) in its funding program "WaX - Wasserextremereignisse".



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
