# Peer review of "Brief Communication: Training of AI-based nowcasting models for rainfall early warning should take into account user requirements"

_EGUsphere, 2024_

## Author Comment (AC1)

**Interactive Discussion: Author Response to Referee #1**

**Brief Communication: Training of AI-based nowcasting models for rainfall early warning should take into account user requirements**

Georgy Ayzel and Maik Heistermann
*EGUSphere,* `doi:10.5194/egusphere-2024-1945`
* * *
**RC:** *Reviewer Comment*,     AR: *Author Response*,     ☐ Manuscript text

Dear referee,

thank you very much for your positive response, and for the time and effort spent to examine the manuscript.

The comments are very useful and will be comprehensively considered in the revised version of the manuscript. Please find a point-by-point reply below.

Thanks again for your willingness to review this manuscript.

Kind regards,
Maik Heistermann
(on behalf of both authors)

**RC:**  *[...] As recognized by the authors, the scientific significance of the result is rather limited, because one would obviously expect the scores of a statistical prediction method to benefit from training with forecast ranges, intensities and weather regimes consistent with those used as verification. Nevertheless, the paper has some educational value for pointing out to the machine learning community that statistical weather prediction models optimized with respect to generic metrics cannot, in general, be expected to perform competitively in terms of more specific metrics.*

**AR:**  We agree that one would "*expect* the scores of a statistical prediction method to benefit from training with forecast ranges, intensities and weather regimes consistent with those used as verification". Still, it is, in our opinion, neither obvious nor self-evident that this expectation would actually materialize, since all models are verified on the same data. A possible outcome could as well have been that, even if trained to predict a specific threshold, the corresponding model could *not* outperform one that was trained on a more generic metric. In fact, this is what we observe for increasingly high thresholds: as predictability deteriorates, all models essentially fail (most evident for the 40 mm/h threshold).

We had pointed out this finding in ll. 154-156 of the preprint:

> These results are in line with our hypothesis that making the training task more specific pays off by a higher predictive skill. One might argue that this result is unsurprising. In our view, though, it is by no means self-evident that the segmentation models could actually capitalize on a more specific training task.

Maybe it would be worth to revisit this statement once more in the conclusions section.

Despite these findings, we entirely agree with the referee that the main point of this paper is of "educational value". This is why we reiterate in ll. 190-192 of the conclusions that

> [...] the aim of our study was not to introduce superior DL architectures or model structures, but to demonstrate how a simplification of the training task can help to improve model skill and to boost the usefulness for specific user groups.

This is exactly why we chose the format of a "brief communication" over a "research paper", with all the consequences e.g. in terms of technical detail of model architectures (see comment below).

**RC:** ***The paper has 11 pages, which seems long for brief communication according to the NHESS website (it indicates a limit of 4 pages).***

AR: As far as we know, the recommended limit of 4 pages for the brief communication format refers to journal pages. As compared to the preprint, the final typeset manuscript will be substantially shorter. Based on our previous experience with this manuscript type (and also considering other brief communications in NHESS), a length of 8 preprint pages (up to the end of the conclusions) plus references corresponds well to the required page limit.

**RC:** ***Not enough information is provided to understand key technical aspects of the study: in particular, there should be a more detailed description of the original RainNet architecture and of the motivation for its changes : what is EfficientNetB4? What is the relevance of LogCosh loss to heavy precipitation? How do you define the threshold on the output to obtain 'a segmentation task'?***

AR: The lower extent of technical detail is owed to the required brevity of the present manuscript type, together with the fact that the main point of the manuscript does not follow from the technical details (see first comment).

Still, we understand that, for parts of the audience, a more detailed technical or methodological description might be desirable. Originally, we had hoped that providing the code repository including its documentation would provide the required technical details to interested readers; however, we agree that an intermediate level of methodological description would be helpful. At the same time, the length of the manuscript is already at the upper limit, and we are convinced that adding such detail in the manuscript would not be helpful to bring across our key message. We therefore suggest to add a supplement to the paper in which we describe in some more detail the abovementioned aspects of e.g. model architectures or loss functions, and refer to this supplementary from the main manuscript.

**RC:** ***False alarms are a key problem when issuing warnings. The dataset used was taken from CatRaRE catalog, which is designed to contain observed extreme events. It means that the training and testing datasets are biased. Thus, the verification will likely underestimate the false alarm frequency, because the dataset excludes cases where the models generate a heavy precipitation forecast and none was observed. A solution would be to compute scores over the entire 2019-2020 testing period. Alternatively the paper should present some proof that the data sampling does not affect false alarm counts. This is the main scientific issue of the paper.***

AR: This is a valid concern. However, even when we focus on situations with heavy rainfall, the model domain of 256 km x 256 km is typically dominated by rainfall accumulations below our thresholds, i.e. by "non-event" grid cells. For our testing data, the frequency of such "non-event" grid cells amounts to 95.73 % for the

threshold of 5 mm in one hour and increases further with increasing precipitation thresholds (10 mm: 98.84 %, 15 mm: 99.56 %, 20 mm: 99.81 %, 25 mm: 99.91 %, 30 mm: 99.96 %, 40 mm: 99.99 %). So, even though the data samples obviously contain enough threshold exceedances for the models to learn, the dominance of "non-event" grid cells prevents overprediction (please also see our response to the referee's below comment), and we would not consider the data sets biased from an application perspective.

In order to address this comment, we will briefly discuss this issue in the manuscript, and add a table in a supplement that provides, for each investigated threshold, the frequencies of events (threshold exceedances) and non-events in the testing data samples.

**RC:** *Significance testing is missing from the results, which is problematic for a paper about statistical prediction. At least, figures 2 and 3 should display some confidence intervals.*

AR: We agree that a measure of significance would make sense in Fig. 2, particularly for very high accumulations for which the skill of all models becomes very low. We will add confidence intervals for the CSI based on a resampling/bootstrapping procedure. For Fig. 3, however, we would prefer to keep the conventional FSS presentation format (please also see our below response to the referee's suggestions on Fig. 3).

**RC:** *The PySteps system is getting old. It would be more convincing to display the scores from a more recent nowcasting system such as DGMR, as a performance baseline.*

AR: Generally, we would agree it would be interesting to include other recent DL-based nowcasting schemes in this analysis. However, the main motivation of the study is not to provide a benchmark experiment among architectures, but to highlight the relevance of considering user requirements in the training task. As we pointed out in the conclusions (ll. 192-193 of the preprint), "this approach should be systematically explored also for recently proposed DL models" (including e.g. DGMR or NowcastNet). Besides, PySteps is still considered a competitive benchmark model representing the conventional approach of optical flow and Lagrangian persistence, and we consider it good practice to include it in the present study. We would therefore prefer to keep the model selection for the benchmark experiment as it is.

**RC:** *The mention of 'user requirements' in the title sounds a bit excessive, because generating rainfall warnings involves other considerations than the choice of accumulation period and threshold. It may be more appropriate to state that the paper demonstrates the sensitivity of nowcast performance to the choice of objective function.*

AR: We understand the referee's reservations with regard to the title. In essence, any objective function is (or should be!) a representation of what a user expects the model to predict. Certainly, user requirements can go beyond the objective function, e.g. in terms of computational efficiency, transparency, comprehensibility and more. However, as we repeatedly addressed in this response, our manuscript aims at reducing the level of technical detail. Our hope is that for the non-technical parts of the audience (as might be the case in NHESS in comparison to other journals), our current title might be more informative, as compared to explicitly referring to objective of loss functions. Besides, we more generally (or unspecifically, if you may) refer to "training" in the title, instead of "objective function", as the training also involves the entire setup including the selection of training data. Of course, the ability of the title to comprehensively reflect the content of the paper is also limited, which is why we explicitly and specifically explain how we define potential user requirements in the context of our study, and make clear that this is only an example.

Altogether, we hope we could explain our motivation of using the term "user requirements" in the title, and we would be glad to keep the title as is due to the given reasons.

**RC:** *Please clarify how can the Jaccard loss can be differentiated, since the Jaccard metric is a ratio of integer*

*success counts.*

AR:    The Jaccard loss is a relaxed, differentiable modification of the Jaccard index, which is a count-based measure similar to the Intersection over Union (IoU) metric and the Critical Success Index (CSI). It describes the ratio between hits and the sum of hits, misses, and false alarms. To address the issue of differentiability, Rahman and Wang (2016) proposed the following relaxation of the Jaccard index:

$$I_{Jaccard} = \frac{\sum_i y_i \hat{y}_i}{\sum_i y_i + \sum_i \hat{y}_i - \sum_i y_i \hat{y}_i}, \tag{1}$$

where:

- $y_i$ is the ground truth binary label for the $i$-th pixel or element.

- $\hat{y}_i$ is the predicted probability for the $i$-th pixel or element.

- $\sum_i y_i$ is the sum of the ground truth binary labels (the count of positive ground truth samples).

- $\sum_i \hat{y}_i$ is the sum of the predicted probabilities (the count of positive predictions).

- $\sum_i y_i \hat{y}_i$ is the sum of the element-wise multiplication of the ground truth and the prediction (the count of true positive samples).

Hence, the Jaccard loss function becomes

$$\mathcal{L}_{\text{Jaccard}} = 1 - I_{Jaccard}. \tag{2}$$

In this way, the gradient of the Jaccard loss function can be computed and integrated into the optimization routine for finding parameters of the neural network.

In the revised version of the paper, we will add the reference to Rahman and Wang (2016).

RC:    *The CSI score should be complemented by some information about the hit rate and false alarm rates (false negatives and false positives), as both are very important for the credibility of warnings.*

AR:    The CSI is already quite a balanced score as it takes into account hits, false alarms and missed events. Given that (even in our samples which present a focus on heavy rainfall situations) the "non-event" grid cells are still dominant, the CSI cannot grow much at the cost of increasing false alarms (see our response to your above comment).

Still, we are willing to compute both hit rates (probability of detection, POD) and false alarm rates (FAR). In order to keep the manuscript brief and follow the requirements of a brief communication, the corresponding figures/tables will be provided in a supplement. We would like to ask for your understanding that we do not already provide the FAR/POD numbers in this interactive discussion. The reason behind this is that we first need to recompute the entire verification since predictions were not stored due to the resulting massive data volumes.

RC:    *typo on line 190 'prediciton'*

AR:    Thanks for spotting the typo which will be fixed in the revised version.

**RC:** *Figure 3 is hard to read in terms of comparison between the systems. Since there is not much information in the dependency on scale, it may be better to present curves of FSS(range) at a fixed scale (say, 20km), instead. It would also facilitate the display of confidence intervals or statistical significance of the FSS differences.*

AR: We do not agree that there is "not much information in the dependency on scale". In our view, demonstrating the dependency on scale is the main motivation of this figure (see also the comments of referee 2). Of course it is obvious that the FSS will increase (or at least not decrease) with increasing scale. The amount of increase is, however, not obvious beforehand, and apparently differs both between models and precipitation thresholds. In our opinion, this figure is not so much about distinguishing exact differences between individual cells, but rather about giving an intuitive and easy to grasp representation of how the FSS varies with model, spatial resolution and precipitation threshold. The actual values printed in the cells are merely a support for those interested in a closer inspection; yet, we do not consider confidence intervals as helpful in the context of this figure (and also not common in this kind of FSS diagrams).

However, if we print the FSS values in the cells, we agree that the legibility could be improved. To this end, we suggest to rearrange the three panels by putting them on top of each other (i.e. 3 x 1 matrix) instead of beside each other (1 x 3 matrix). That way, the plot could be larger, so that the font size of the labels can be increased, too.

**References**

Rahman, M. A. and Wang, Y.: Optimizing Intersection-Over-Union in Deep Neural Networks for Image Segmentation, in: Advances in Visual Computing, edited by Bebis, G., Boyle, R., Parvin, B., Koracin, D., Porikli, F., Skaff, S., Entezari, A., Min, J., Iwai, D., Sadagic, A., Scheidegger, C., and Isenberg, T., pp. 234–244, Springer International Publishing, Cham, ISBN 978-3-319-50835-1, 2016.

---

## Author Comment (AC2)

**Interactive Discussion: Author Response to Referee #2**

**Brief Communication: Training of AI-based now-casting models for rainfall early warning should take into account user requirements**

Georgy Ayzel and Maik Heistermann
*EGUSphere,* `doi:10.5194/egusphere-2024-1945`
* * *
**RC:** *Reviewer Comment*,     AR: *Author Response*,     ☐ Manuscript text

Dear Prof Uijlenhoet,

thank you very much for your extensive and constructive response, and for the time and effort spent to examine the manuscript.

The comments are very useful and will be considered in the revised version of the manuscript.

Your comments also provide a venue for a discussion that, in many parts, goes beyond the limited scope of this brief communication. All the more we appreciate the opportunity of this public discussion format.

Please find a point-by-point reply below. Although this reply contains some redundancies (as questions of scale were repeatedly raised), we still decided to stick with the point-by-point approach for the sake of transparency and traceability. We apologize in advance for the resulting length of the response.

Thanks again for your willingness to review this manuscript.

Kind regards,
Maik Heistermann
(on behalf of both authors)

**RC:** *[...] Overall, the paper makes a valid point, is well-written and provides some clear and convincing illustrations. As such, it provides a timely and relevant perspective on the state of the art of rainfall nowcasting for the readers of NHESS.*

AR: Thanks for the positive response.

**RC:** *As an example of a possible user requirement, the authors consider "the exceedance of thresholds (instead of numerical values)" (l. 4-5). They claim that "this has been rarely attempted so far" (l. 31-32). Note, however, that the use of rainfall thresholds is actually common practice in the area of "flash flood guidance" (e.g. Georgakakos et al., 2022).*

AR: Most likely, this might be a misunderstanding. The claim that "this has been rarely attempted before" refers to the training of DL-based nowcasting models in order to predict the exceedance of user-relevant precipitation thresholds. So this claim should be read in the context of the full paragraph (ll. 29-33 of the preprint):

> Our hypothesis is that DL models have difficulties in detecting generalizable patterns in case they are trained to predict a wide range of precipitation intensities and depths. We further hypothesize that this issue could be addressed by tailoring the training task and procedure more towards user-relevant precipitation thresholds. It is surprising that this has been rarely attempted so far (with the exception of Leinonen et al., 2023) – since the possibility to train DL models for solving specific tasks is one of their inherent strengths.

We will slightly adjust the text to make it less prone to misunderstandings ("[...] could be addressed by tailoring the training task and procedure more towards the prediction of whether user-relevant precipitation thresholds will be exceeded [...]")

**RC:** *"As an example, we predict the cumulative precipitation of the next hour" (l. 4). Why this particular duration? What about longer lead times (probably necessitating the connection to numerical weather prediction models), which may also be of interest for certain applications?*

AR: The duration of one hour was, as stated, chosen as an example. Still, the referee is right: for many applications, longer lead times would be desirable. Given, however, the existing evidence that the predictability of heavy rainfall systems, specifically in convective situations, dramatically deteriorates in the first hour, we focused our efforts on that first hour. That does certainly not exclude the possibility of future progress, if we can e.g. combine convection-permitting atmospheric models with deep neural networks (see comment below). However, that topic is beyond the scope of this brief communication.

Towards the other end (i.e. addressing accumulations *shorter* than 60 minutes), many stakeholders / institutions / protocols in the context of early warning refer to *one hour* as the shortest duration. In Germany, for instance, the official DWD warning levels only refer to durations of one hour *or* six hours; this is similar for flash flood guidance systems, as the referee mentioned himself. Certainly, there are users who require nowcasts at a higher temporal resolution (i.e. shorter accumulation intervals), for instance for advanced urban drainage control and/or for very small catchments. From a hydrological perspective, the required temporal resolution results from a catchment's concentration time which in turn is a function of size and surface properties such as topography and land use.

Summing up, we look at precipitation accumulation over the next hour as a compromise between maximum skillful lead times and minimum durations specified by many warning protocols, based on the fact that it is at these durations where severe hydrological impacts typically begin to unfold (except for e.g. very small catchments). In the revised version of the manuscript, we will *very briefly* explicate these considerations around ll. 36-40 of the preprint in order to justify our choice. In our view, however, going into the details of spatial and temporal scale issues in hydrology and precipitation forecasting would - interesting and relevant as they may be - not serve the purpose of our paper well.

**RC:** *Concerning point 1 (l. 36-40): What can be said for the required temporal resolution from the perspective of user requirements also holds for the spatial scale of application, e.g. an urban area, river catchment, etc. However, whereas the temporal aspect is considered here in quite some detail, the spatial scale of application (referring to the total size of the domain over which the forecast is produced and evaluated) appears to be neglected here.*

AR: We agree that our focus is on the role of temporal resolution, or, more specifically, on replacing the five minute resolution by a one hour aggregation. We are aware that user requirements might as well vary with regard to the spatial scale.

The referee, in his comment, refers to spatial scale as the extent (or total size) of the model domain. While the extent is an important aspect of scale, the spatial *resolution*, however, strikes us as more important in the nowcasting context (and would also be the analogue to the temporal resolution). While we agree that, in our benchmark experiment, we only consider the aspect of temporal resolution/aggregation, we tend to disagree that "[...] the spatial scale [...] appears to be neglected [...]" in our study: with the Fractions Skill Score (FSS, Fig. 3 of the preprint), we have touched upon the issue of spatial resolution in the sense of an acceptable displacement error:

In ll. 157-159 of the preprint, we wrote

> It is well known that, particularly in convective situations, nowcasting models struggle to provide skillful forecasts at the kilometre-scale. The Fractions Skill Score (FSS) quantifies the model skill when we relax this requirement, i.e. when we allow an increasing level of displacement error.

Then, in ll. 164-167, we wrote that

> the FSS demonstrates an additional dimension along which the training task for precipitation nowcasts could be relaxed in case users do not require a kilometre-scale resolution. Although RainNet2024-S is already superior at all spatial scales, its skill might well be pushed further if directly trained for a specific spatial scale, or, in other words, if the displacement error acceptable by the user were directly considered in model training.

and, finally, in ll. 194-196:

> There are various conceivable dimensions along which user preferences might find their way into model training, e.g. by specifying precipitation thresholds, spatial and temporal resolution, or preferences towards deterministic versus probabilistic forecasts.

This discussion is essentially related to the question at which resolution a model is evaluated (or trained), and we entirely agree that this is a matter of user preference: if a nowcast is to be applied in a catchment of say 400 km² size, a displacement error of a few kilometres becomes acceptable to the user, and a DL model could even be trained to directly predict the areal average rainfall accumulations for larger spatial units, let it be grid cells or specific catchments (although the latter might cause issues in terms of spatial transferability of the model).

In the context of the manuscript under discussion, we would prefer not to delve much deeper than we already did in the discussion of the FSS results and scale issues. Around ll. 164-167, however, we suggest to briefly refer to Lin et al. (2024) who confirmed a strong dependency of predictive skill on the size of the prediction target (in this case cities).

In addition, we fully agree that our use of the term "spatial scale" is partly imprecise in the preprint, and we will revise the manuscript to always be clear about whether we refer to the spatial extent of the model domain or the spatial resolution.

RC:  *"on the same domain" (l.52-53): See previous remark concerning the spatial scale (i.e. domain) of application. In addition, for catchment hydrological applications, relevant spatial and temporal scales are related to each other (see e.g. Berne et al., 2004).*

**AR:** We agree, and we are aware. Please also refer to the above and below comments related to spatial scale. For us, the main subject of spatial scale is the resolution of the forecast, as already noted. *Maybe* the extent of the model domain has an effect on model skill in the case of DL models (maybe less so for conventional tracking and extrapolation models, but still possible), but this has, to our knowledge, not yet been investigated in depth. For the user, however, it is more important at which resolution to interpret the forecast. If the model domain had an extent of 1024 km x 1024 km and a resolution of 1 km x 1 km, some users in some small city might interpret the results at this native resolution while others would e.g. rather compute the areal average rainfall over a river catchment. The question whether there is some inherent trade-off between extent and resolution when if comes to deep learning is a more fundamental one, and our study does not provide any basis for a detailed discussion of that issue with regard to nowcasting and hydrological applications.

**RC:** *"heavy rainfall objects (l.66): How are heavy rainfall "objects" defined exactly?*

**AR:** The term "objects" might be a bit misleading. The fundamental idea of the CatRaRE catalog is to detect rainfall events that show some coherency in space and time. For the sake of the catalog, this coherency is important since specific event metrics (e.g. duration, extremity) are then derived for these events. In the context of our study, the catalog just allowed us to conveniently pinpoint times and regions with relevant rainfall activities. By spanning a domain 256 km x 256 km around these events, however, we typically include large areas without heavy rainfall which is important in order for the model to learn how to avoid false alarms (see also comment of referee 1 and below comment of this referee). The exact approach how to detect and delineate the catalogued events is described in detail by Lengfeld et al. (2021). Instead of describing it in the brief communication, we prefer to refer more explicitly to that source. In the revised manuscript, we will hence replace the term "objects" by "events", and mention explicitly that the underlying detection approach is described in Lengfeld et al. (2021).

**RC:** *"a spatial domain of 256 x 256 km" (l.80): This seems a rather arbitrary spatial scale (domain), except for the fact that it is $2^8$ km x $2^8$ km. How does this spatial domain "match" with the hourly time step that was chosen from a (rainfall-runoff) process perspective?*

**AR:** As pointed out above, the *extent* of the spatial domain (here: 256 km x 256 km) is maybe less important than the *resolution* at which the forecast is generated (here: 1 km x 1 km). Users can also average (aggregate) in space, based on their preferences, but as discussed with regard to the referee's above comment, future research might also use a different resolution for model training (or train the model directly to predict average areal rainfall for specific catchments). In our case study, we kept the standard kilometre resolution which is also the original spatial resolution of the radar composite.

The decision about the spatial *extent*, however, is rather governed by technical aspects. The upper end of the spatial extent is set by the extent of the radar composite. The real-time composite provided by DWD has an extent of 900 km x 900 km. As the efficient training of deep neural networks on GPUs requires grid dimensions that correspond to a power of 2, possible choices in our setup would have been a domain size of 1024 km (which would require substantial padding which is inefficient), 512 km or 256 km.

In DL-based nowcasting, a domain size of 256 km x 256 km has been increasingly used by recent studies for model training, including the paper of Zhang et al. (2023) on NowcastNet. This is also owed to the fact that a lot of developments have been done with regard to processing images of that size (e.g. computational improvement of tensor operations, available architectures and backbone weights).

Limiting our analysis to a lead time of one hour, a domain size of 256 km was considered large enough to avoid the dominance of edge effects (when incoming rain from luv is not yet within the forecast domain).

When you aim at a larger spatial coverage of a nowcast (e.g. Germany-wide), a larger application domain is

straightforward to implement by merging nowcasts from overlapping 256 km x 256 km domains to a larger spatial domain. However, that is rather are deployment issue.

After all, preferring 256 km over 512 km is still a somewhat arbitrary decision, although one might hope that a smaller domain might support the model in better learning small-scale precipitation dynamics in convective situations (which we were focusing on in this study), see also the response to the above comment.

Having said all that, we think that it will not serve the audience to elaborate on the technical justification of the spatial domain extent in the scope of this brief communication. With regard to the spatial resolution, we hope that the slight extension of the manuscript, as outlined in our response to the above comment, will address the referee's concern of an imbalance between treatment of temporal and spatial scales.

**RC:**  *"at a duration of six hours or less" (l.111): Why this particular range of durations? Was this choice the result of an interaction with stakeholders?*

 AR:  This choice was mainly governed by the aim to focus the model training on particularly challenging precipitation situations, i.e. on convective situations with heavy rainfall and strong spatio-temporal dynamics. Again, there is some level of arbitrariness involved: given that our lead time was limited to one hour, we could also have restricted our choice to events of one hour duration. But of course that perspective would have been too narrow, since events that have a longer duration can also have relevant short-term dynamics which could contribute to a successful model training. Enhancing the duration criterion towards much longer durations would, however, include events that have a larger spatial extent and slower dynamics which would simply increase model skill in testing (Lin et al., 2024), but not add much to learning the prediction of volatile convective events. In the end, the entire benchmark experiment could have been designed much more comprehensively by including different event lengths, lead times, spatial resolutions and temporal aggregations...but that would have been a different study and a different manuscript format (please also see our response to the below comment in that regard).

So with regard to this specific comment, we consider the previous level of justification, as provided in ll. 110-114 of the preprint, as sufficient:

> For model training and testing, we selected [...] events between 2001 and 2020 which were most extreme at a duration of six hours or less [...]. That way, we created a particularly challenging benchmark environment, since we not only focus our analysis on extreme precipitation events, but specifically on events with a relatively short duration.

**RC:**  *"the kind of events [...] early warning context" (l.115-116): Have the authors interacted with stakeholders to define their (subjective choices of) lead time, rainfall thresholds, spatial resolution, scale of application (domain) and durations?*

 AR:  In fact, we did. In an ongoing research project (`https://www.uni-potsdam.de/de/inno-maus/`), we had a discussion with different stakeholders at the municipal and state levels, in which emerged a support of the temporal aggregation of one hour, as well as of keeping the spatial resolution at one kilometre. As already elaborated above, the choice of the 1h lead time is owed to the predictability (while stakeholders always ask for longer lead times). As for the thresholds, we highlighted, in Fig. 2, the official DWD warning levels as a basis of user requirements, but also included other thresholds in the investigation. The upper limit of 40 mm/h is a result of that fact that predictability is obviously lost at this level, at least for our models.

We think that the main justification of these features is mostly already contained in the manuscript, or in the planned revisions as outlined in our above responses.

**RC:** *Fig.2: It would be interesting and relevant to see the corresponding curves for 2-hour and 3-hour rainfall accumulations (why not up to 6 hours, the maximum duration selected) and different domain sizes, going from 256 km x 256 km down to 2 km x 2 km (following Lin et al., 2024).*

**AR:** We can just agree with the referee: yes, it would be interesting. Still, it would be a different study, in terms of a training and verification experiment that comprehensively investigates the predictive skill of deep learning models subject to spatial and temporal scale. Certainly, this would not be a brief communication, but a research paper. We hope that the referee is not annoyed if we emphasize, again and again, that this paper has a very distinct point to make, and that we make this point only by example, but certainly hoping to stimulate prospective research.

**RC:** *Fig.3, "spatial scale": Is this "scale" or is this "resolution", keeping the same 256 km x 256 km spatial domain? Ultimately, the domain size over which the statistics are calculated (related to the area of application of the nowcast) is also relevant from a practical (end-user / stakeholder) perspective.*

**AR:** Fig. 3 actually refers to spatial resolution. We will be more specific with regard to the terms of spatial resolution and extent in the revised manuscript, also see our above and below comments. From the user perspective, however, the model domain (extent) is not of key interest (in our opinion).

**RC:** *"the RainNet2024-S models clearly outperform all competitors across all precipitation thresholds" (l.152): This is certainly a nice performance of the presented model for the selected heavy rainfall events, but a question one may ask is how robust these results are in terms of false alarms?*

**AR:** This issue was raised by referee 1, too. The CSI is already quite a balanced score as it takes into account hits, false alarms and missed events. As already pointed out to referee 1, the frequency of "non-event" grid cells in our testing data amounts to 95.73 % for the threshold of 5 mm in one hour, and increases further with increasing precipitation thresholds (10 mm: 98.84 %, 15 mm: 99.56 %, 20 mm: 99.81 %, 25 mm: 99.91 %, 30 mm: 99.96 %, 40 mm: 99.99 %). So, even though the data samples obviously contain enough threshold exceedances for the models to learn, the dominance of "non-event" grid cells prevents overprediction, and the CSI cannot grow much at the cost of increasing false alarms.

Still, we are willing to compute both hit rates (probability of detection, POD) and false alarm rates (FAR). In order to keep the manuscript brief and follow the requirements of a brief communication, the corresponding figures/tables will be provided in a supplement. We would like to ask for your understanding that we do not already provide the FAR/POD numbers in this interactive discussion. The reason behind this is that we first need to recompute the entire verification since predictions were not stored due to the resulting massive data volumes.

**RC:** *"or fields from numerical weather prediction models" (l.190): Including NWP forecasts could hopefully further increase the skillful lead time. What is the authors' perspective on merging ML-based radar rainfall nowcasts with NWP forecasts for seamless prediction up to longer lead times than just a few hours?*

**AR:** In fact, we had already pointed out the possibility of including NWP fields in our 2020 RainNet paper (Ayzel et al. 2020), but have not embarked on that journey since then. And not so many others, either, with the exception of e.g. Leinonen et al. (2023), or, more recently, Kim et al. (2024), both already cited in the preprint. As the referee probably knows better than us, the efforts to merge / blend / fuse NWP forecasts with radar-based nowcasts are ongoing since decades, from simple time-dependent weighting to more advanced techniques (scale- and skill dependent blending, ensemble techniques, assimilation, ...). Recent developments achieved skills that were, for any given lead time, at least not worse than either the nowcast or the NWP model (see e.g. Nerini, 2019; Imhoff et al., 2023). But of course, the blending approach will not be able to

magically boost predictive skill for any given lead time around a few hours time.

Of course, hopes are up that DL might allow for such a boost, by actually being able to learn (or "reveal") intrinsic relationships between NWP patterns (not necessarily forecasts, could also just be fields such as CAPE, vorticity, or moisture convergence at forecast time) and imminent dynamics as observed by weather radar. Leinonen et al. (who used CAPE from the COSMO model) found in their specific study that "radar is the most important source, followed by the lightning and satellite data. NWP is less important but nevertheless clearly beneficial." While this does not yet sound like the hoped for "boost", it is clearly a progress. This is similar in Kim et al. (2024) who used the ERA5 fields "total column water vapour" and "divergence at 925 hPa" as additional inputs, claiming that "applying the proposed loss function and using the divergence at 925 hPa as an additional input, the deep-learning model [...] obtained the highest ETS values". The actual margins at which this setup actually outperformed its competitors without additional NWP inputs are quite narrow, and the absolute skill even in the first hour is quite low for high rainfall intensities (ETS smaller than 0.05 for e.g. 20 mm/h). For lead times longer than one hour (the study investigated lead times of up to six hours), the skill drops sharply (ETS virtually zero for a threshold of 20 mm/h and lead times > 1 hour),

So, while progress in that regard might be slow, things are generally moving fast in the AI department. Still, the intrinsic limits of predictability, as governed by the hot temper of convective systems, will be difficult to push. Hence the quantification of predictive uncertainty will remain paramount, also for DL-based nowcasts.

While we were glad to share our perspective with the referee in the context of this response, we feel that any such discussion would be off topic in the context of our brief communication, and we hope that the referee agrees.

**RC:** *l.58: "2018a,b" instead of "2018b,a"?*

 AR: Will be implemented.

**RC:** *l.80: "256 km x 256 km" rather than "256 x 256 km".*

 AR: Will be implemented.

**RC:** *Fig.1: "YW" presumably refers to "the RADKLIM_YW_2017.002 dataset (Winterrath et al., 2018b,a)" (l.58). However, this should be more clearly indicated (2x). Also, the authors may want to expand the caption of this figure into one that is slightly more informative.*

 AR: We will expand the figure caption, and also use the figure caption to clarify the reference of "YW".

**RC:** *"kilometre-scale resolution" (l.165): Here, the term "resolution" is used, where previously (including in the caption of Fig. 3) the authors use "scale". It would be appropriate to clearly distinguish one from the other.*

 AR: We fully agree. Given also the above comments, we will be more specific with regard to terminology on spatial scale. We will revise the manuscript to avoid the unspecific use of the term "spatial scale", and instead differentiate between spatial extent (of the model domain) and spatial resolution.

**RC:** *"spatial scale" (l.186): See previous remark.*

 AR: See previous response.

**RC:** *l.253: "Brendel" instead of "Brend".*

 AR: Will be corrected.

**References**

Berne, A., Delrieu, G., Creutin, J.-D., and Obled, C.: Temporal and spatial resolution of rainfall measurements required for urban hydrology, Journal of Hydrology, 299, 166–179, https://doi.org/https://doi.org/10.1016/j. jhydrol.2004.08.002, urban Hydrology, 2004.

Georgakakos, K. P., Modrick, T. M., Shamir, E., Campbell, R., Cheng, Z., Jubach, R., Sperfslage, J. A., Spencer, C. R., and Banks, R.: The Flash Flood Guidance System Implementation Worldwide: A Successful Multidecadal Research-to-Operations Effort, Bulletin of the American Meteorological Society, 103, E665 – E679, https://doi.org/10.1175/BAMS-D-20-0241.1, 2022.

Imhoff, R. O., De Cruz, L., Dewettinck, W., Brauer, C. C., Uijlenhoet, R., van Heeringen, K.-J., Velasco-Forero, C., Nerini, D., Van Ginderachter, M., and Weerts, A. H.: Scale-dependent blending of ensemble rainfall nowcasts and numerical weather prediction in the open-source pysteps library, Quarterly Journal of the Royal Meteorological Society, 149, 1335–1364, https://doi.org/https://doi.org/10.1002/qj.4461, 2023.

Kim, W., Jeong, C.-H., and Kim, S.: Improvements in deep learning-based precipitation nowcasting using major atmospheric factors with radar rain rate, Computers Geosciences, 184, 105 529, https://doi.org/https://doi.org/10.1016/j.cageo.2024.105529, 2024.

Leinonen, J., Hamann, U., Sideris, I. V., and Germann, U.: Thunderstorm Nowcasting With Deep Learning: A Multi-Hazard Data Fusion Model, Geophysical Research Letters, 50, e2022GL101 626, https://doi.org/https://doi.org/10.1029/2022GL101626, e2022GL101626 2022GL101626, 2023.

Lengfeld, K., Walawender, E., Winterrath, T., and Becker, A.: CatRaRE: A Catalogue of radar-based heavy rainfall events in Germany derived from 20 years of data, Meteorologische Zeitschrift, pp. 469–487, https://doi.org//10.1127/metz/2021/1088, publisher: Schweizerbart'sche Verlagsbuchhandlung, 2021.

Lin, G.-S., Imhoff, R., Schleiss, M., and Uijlenhoet, R.: Nowcasting of High-Intensity Rainfall for Urban Applications in the Netherlands, Journal of Hydrometeorology, 25, 653 – 672, https://doi.org/10.1175/JHM-D-23-0194.1, 2024.

Nerini, D.: Ensemble precipitation nowcasting: limits to prediction, localization and seamless blending, Phd thesis, ETH Zurich, Zurich, Switzerland, https://doi.org/10.3929/ethz-b-000391932, available at https://doi.org/10.3929/ethz-b-000391932, 2019.

Zhang, Y., Long, M., Chen, K., Xing, L., Jin, R., Jordan, M. I., and Wang, J.: Skilful nowcasting of extreme precipitation with NowcastNet, Nature, 619, 526–532, https://doi.org/10.1038/s41586-023-06184-4, number: 7970 Publisher: Nature Publishing Group, 2023.